# First-in-Human Integrated Use of a Dedicated Microsurgical Robot with a 4K 3D Exoscope: The Future of Microsurgery

**DOI:** 10.3390/life13030692

**Published:** 2023-03-03

**Authors:** Tom J. M. van Mulken, Shan S. Qiu, Yasmine Jonis, Jairo J. A. Profar, Taco J. Blokhuis, Jan Geurts, Rutger M. Schols, René R. W. J. van der Hulst

**Affiliations:** 1Department of Plastic, Reconstructive and Hand Surgery, Maastricht University Medical Center, P. Debeyelaan 25, 6229 HX Maastricht, The Netherlands; 2Department of Trauma Surgery, Maastricht University Medical Center, Maastricht, P. Debeyelaan 25, 6229 HX Maastricht, The Netherlands; 3Department of Orthopedic Surgery, Maastricht University Medical Center, P. Debeyelaan 25, 6229 HX Maastricht, The Netherlands

**Keywords:** microsurgical robot, exoscope, microsurgery, plastic and reconstructive surgery, innovation

## Abstract

Innovative techniques can help overcome the limitations of the human body. Operating on very small structures requires adequate vision of the surgical field and precise movements of sophisticated instruments. Both the human eye and hand are limited when performing microsurgery. Conventional microsurgery uses operation microscopes to enhance the visualization of very small structures. Evolving technology of high-definition 3D cameras provides the opportunity to replace conventional operation microscopes, thereby improving ergonomics for surgeons. This leaves the human hand as a limiting factor in microsurgery. A dedicated robot for microsurgery has been developed to overcome this limitation and enhance the precision and stability of the surgeons’ hands. We present the first-in-human case in reconstructive microsurgery where both technologies are integrated using a dedicated microsurgical robot in combination with a 4K 3D exoscope.

## 1. Introduction

Microsurgery signifies a part of surgery that requires microscopes and surgical loupes and is used by several surgical specialties. In reconstructive surgery, microsurgery is mainly used for anastomosis of small blood vessels, nerves, and lymphatics. This allows for vascularized free tissue transfer to reconstruct defects, innervated free muscle transfer to reconstruct function, lymphatic surgery to treat lymphedema, and the repair of vital tiny structures after trauma. Microsurgery requires a high-quality magnification of the surgical field and controlled delicate motions of the operating surgeon.

As the human hand and eye are limited, innovative techniques can help overcome these limitations and enhance the surgeon’s technical abilities. A robotic system [1] can allow for tremor filtration and motion scaling to improve the precision of the human hands. In the literature, several robotic systems have been evaluated for microsurgery. Besides improved dexterity, other positive factors of robotic assistance that have been described are fast (steep) learning curves and ergonomics [2,3].

A 3D exoscope [4] can provide a high-quality magnification while preserving the ergonomic posture of the operating surgeon, which is often suboptimal when using conventional operation microscopes or surgical loupes.

This article describes our early clinical experience in integrating a dedicated microsurgical robot with a 4K 3D exoscope.

## 2. Materials and Methods

### 2.1. Microsurgical Robot

The MUSA robot (MicroSure B.V., Son, The Netherlands) is the first dedicated robotic platform for open microsurgery (see Figure 1). It was designed and developed in a conjoined effort of the Maastricht University Medical Center (Maastricht) and the Eindhoven Technical University (Eindhoven) in The Netherlands.

The system consists of two robotic arms that are attached to a ring positioned above the surgical field. The robotic arms can be loaded with genuine (super)microsurgical instruments and copy the movements of two master manipulators that are controlled by the operating surgeon.

By tremor filtration and motion scaling, the movements of the operating surgeon are transferred into precise and controlled micro movements. Preclinical and clinical pilot studies have confirmed the safety and feasibility performing microsurgical anastomoses, and even super-microsurgical anastomoses of vessels between 0.3 and 0.8 mm in diameter [1,5,6]. The system can be combined with any operating microscope or exoscope.

### 2.2. Exoscope

The ORBEYE 4K 3D system (Olympus Inc., Tokyo, Japan) is an exoscope (see Figure 2). The dual-optics camera system is attached to a flexible and freely maneuverable arm that is connected to the processor containing the base of the system. An LED light source is transmitted via fiber optics to the camera head. The image is projected to a 31-inch 4K 3D monitor with a 4160 × 2160-pixel resolution. The combination of a 1:6 optical zoom, 2× digital zoom and magnification of the large screen can provide up to 26× magnification. The 3D image is achieved by wearing polarized 3D glasses in combination with the system. A second smaller screen is available for the assisting surgeon and is placed directly behind the operating surgeon, and an additional 2D output can be forwarded to other screens in the operation room for other personnel [4].

### 2.3. Clinical Case

In March 2022, a 72-year-old male patient was referred to our multidisciplinary nonunion clinic with a nonunion of the distal tibia and fibula after a high-energy Gustillo grade 2 open comminuted fracture of his left leg in August 2021. Initially the injury was treated with debridement of the wound and an external fixation, followed by definitive fixation with a plate. Because of delayed union, an additional cancellous bone grafting procedure was performed in November 2021, but no fracture healing occurred (see Figure 3).

He had no other relevant medical history or medication and is a non-smoker. Clinical examination revealed multiple scars on the anterior part of the lower extremity with a small crust in the distal part of the scar, an indicator for a low-grade infection. Additional radiologic examination by Positron Emission Tomography/Computed Tomography (PET-CT) scan confirmed a comminuted crural fracture nonunion without signs of osteomyelitis.

The treatment plan consisted of a 2-staged Masquelet approach [7,8]. The Masquelet technique is a 2-staged procedure used to address segmental bone defect reconstruction. The first operation is focused on the debridement of all infected and necrotic tissue to achieve a clean cavity, which is temporarily filled with a cement block and covered with well-vascularized soft tissue coverage. This is usually performed with a free tissue transfer, harvesting an autologous soft tissue flap and reconnecting the vascular in- and outflow of this flap using microsurgical techniques.

This first stage induces the formation of a richly vascularized membrane around the cement spacer of the bone defect. After 6–8 weeks, the second stage consists of removal of the cement spacer and filling of the bone cavity with bone graft, which is then surrounded by this well-vascularized membrane and soft tissue envelope, promoting revascularization of the bone graft.

We describe the first clinical case of integrated use of an exoscope (ORBEYE, Olympus Inc., Tokyo, Japan) and dedicated microsurgical robot (MUSA, MicroSure B.V., The Netherlands) for the microvascular reconstruction of soft tissue with a free flap.

## 3. Results

### Surgical Procedures

Stage 1 of the Masquelet procedure (21 June 2022): Debridement of the necrotic soft tissue and bone elements, removal of the plate and screws and insertion of a T2 tibia nail (Stryker, Kalamazoo, MI, USA), insertion of an antibiotic-loaded cement spacer (Refobacin Bone Cement R, Zimmer- Biomet, Warsaw, IN, USA) and reconstruction of the soft-tissue defect with a free vascularized Antero-Lateral-Thigh (ALT) flap from the left upper leg were carried out.

The patient was placed in a supine position under general anesthesia. At induction, prophylactic antibiotics (intravenous amoxicillin/clavulanic acid) were provided. The first part of the procedure was performed by the orthopedic and trauma surgeons for debridement of all avascular soft and bony tissue, removal of plate and screws, subsequent bone fixation using a T2 tibia nail, and placement of a cement spacer (see Figure 4 for the intra-operative X-ray). The plastic surgeons then performed the reconstruction of the soft tissue defect using a free Antero-Lateral-Thigh (ALT) flap from the ipsilateral leg. This is a fasciocutaneous flap consisting of skin, subcutaneous tissue and muscle fascia of the antero-lateral side of the upper leg. Using surgical loupes, both the donor and acceptor sites were dissected. The ALT flap was raised on one musculocutaneous perforator. In parallel, the posterior tibial artery and vein were dissected on the lower leg as recipient vessels for this free ALT flap.

The MUSA robot and ORBEYE exoscope were draped and installed above the surgical field (see Figure 5). The MUSA robot was loaded with genuine microsurgical instruments (micro forceps and needle holder of S&T, Synovis Inc., Birmingham, AL, USA) and was activated using a foot pedal providing tremor filtration and motion scaling for the operating surgeon.

The ORBEYE system was combined with 3D-polarized glasses. The operating surgeon controlled the MUSA robot while looking at the large 31-inch screen, the assisting surgeon was positioned in a conventional way opposite the operating surgeon while looking at the smaller screen using 3D glasses. Using 9.0 sutures (Ethicon, Johnson and Johnson, New Brunswick, NJ, USA), an end-to-end anastomosis was created of the posterior tibial artery to the pedicle of the ALT flap. A coupler system (FLOW COUPLER, Synovis USA) was used to create an end-to-end anastomosis between the posterior tibial vein and the pedicle of the ALT flap.

After finishing the microsurgical anastomoses, both the MUSA robot and ORBEYE system were moved away from the surgical field and the well-perfused flap was secured to the defect using conventional resorbable sutures.

Prophylactic antibiotics (intravenous amoxicillin/clavulanic acid) were continued. Anticoagulant prophylaxis was provided until 6 weeks after surgery.

From postoperative day 7, the patient was allowed free mobilization, i.e., unlimited dangling of the left leg. He showed an uneventful recovery and was discharged home on postoperative day 14. At discharge, antibiotics were stopped as intraoperative cultures remained negative.

Stage two of the Masquelet procedure (15 August 2022): Partial elevation of the ALT flap followed by removal of the cement spacer and insertion of an autologous bone graft that was obtained from the ipsilateral femur using the Reamer–Irrigator–Aspirator (RIA) system (Johnson and Johnson, USA), combined with autologous mesenchymal stem cells that were harvested from a bone marrow punction at the ipsilateral Iliac crest using the Angel centrifuge system (Arthrex, Naples, FL, USA) (RIA/BMAC-procedure) and replacement of the T2 tibia nail (Stryker, Kalamazoo, MI, USA), were carried out.

The patient returned to our multidisciplinary outpatient clinic for follow-ups, reporting a viable, nicely healed ALT flap with no signs of infection and progressing consolidation of the bone defect (21 December 2022; see Figure 6 and Figure 7). Proper knee and ankle joint function was confirmed while weight bearing/walking was further accomplished in combination with physiotherapy guidance.

## 4. Discussion

We present the first-in-human integrated use of a dedicated microsurgical robot and a 4K 3D exoscope for microsurgical free tissue transfer, embodying the future of reconstructive microsurgery. These two innovative surgical techniques help us to overcome limitations of the human eye and hand, which are especially challenging in reconstructive microsurgery.

Operation microscopes are currently the gold standard for optical magnification in microsurgery. They provide a high-quality, three-dimensional magnified view of the operation field. A limitation of operation microscopes is that the operating surgeon needs to look through the oculars while the microscope is positioned above the operation field.

This often requires a suboptimal ergonomic position for the surgeon, who is fixed to this position for many hours, concentrating on the delicate and small structures. Many microsurgeons are being confronted with musculoskeletal disorders of the shoulders, cervical area and lumbar spine in the long term [9,10,11,12].

A great advantage of the incorporation of exoscope imaging in reconstructive microsurgery is the enhanced ergonomics for the operating surgeon. The compact camera-system can be placed over the operation field and allows the surgeon to maintain optimal ergonomics while looking through 3D glasses. In addition to this enhanced ergonomic position, the external monitor also decouples the surgeons’ eyes from the microscope oculars, eliminating the need to maintain a fixed position for a prolonged time [13,14].

Robotic assistance has the potential to provide superhuman precision in microsurgery. Motion scaling can transfer relatively large movements of the operating surgeon to very small movements of the robotic arm that is loaded with superfine microsurgical instruments. In addition, the software can filter out any tremor of the surgeon’s hand.

The limitations of robotic systems are time to docking the robot and costs [15].

The MUSA robot is the first robot that is specifically designed for microsurgical operations. The system can be loaded with genuine microsurgical instruments using 3D-printed adapters and it can be combined with any microscope or exoscope system.

Preclinical and clinical studies have confirmed the feasibility of robotic microsurgery using the MUSA robot [1,5]. Super microsurgical operations (defined as anastomosis of vessels with a diameter between 0.3–0.8 mm) are currently considered as the frontier of microsurgery. A prospective randomized pilot trial on lymphedema reconstructions has confirmed the safety and feasibility of the MUSA system in these super microsurgical operations [6,16].

The first clinical combined use of a dedicated microsurgical robot and an exoscope shows the feasibility of its integrated application. Enhanced precision through robotic assistance is combined with a 4K 3D view while preserving the ergonomics of the operating surgeons and staff. In our opinion, the evolution of the field of (super)microsurgery will be based on the combination of these two innovative surgical techniques. Robotic platforms will continue to improve, and new platforms will be introduced. In parallel the quality and size of exoscope camara systems will continue to evolve, and in our opinion, they will ultimately replace the current operation microscopes.

In conclusion, the combined use of a dedicated microsurgical robot and exoscope imaging results in enhanced visualization, precision, and ergonomics for (micro)surgeons. These benefits will result in improved patient outcomes and also decrease the number of musculoskeletal disorders for medical staff. We should continue to explore and develop these potential benefits in a structured and judicious way considering safety, costs and limitations and continue the path to better care for our current and future patients.

## Figures and Tables

**Figure 1 life-13-00692-f001:**
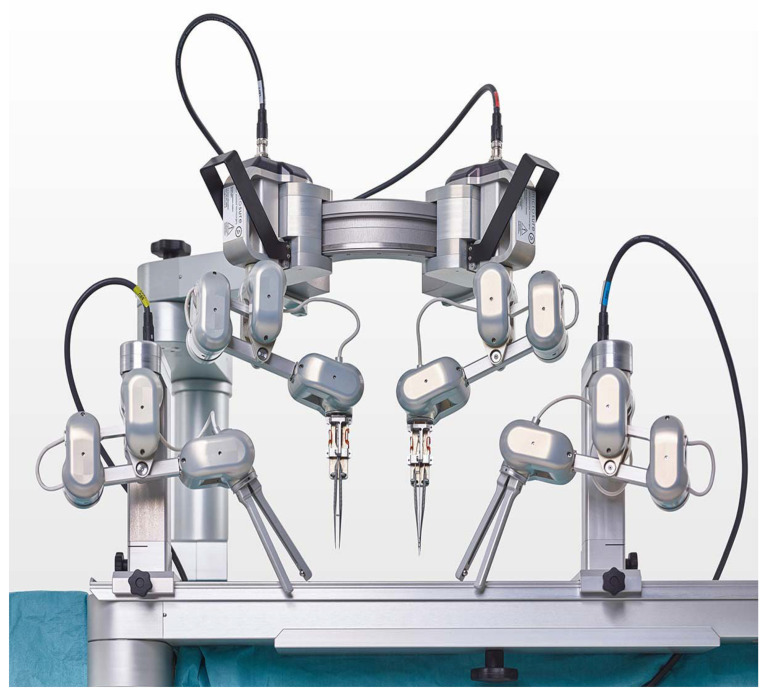
MUSA robot (Microsure B.V., Son, The Netherlands), a dedicated microsurgical robot.

**Figure 2 life-13-00692-f002:**
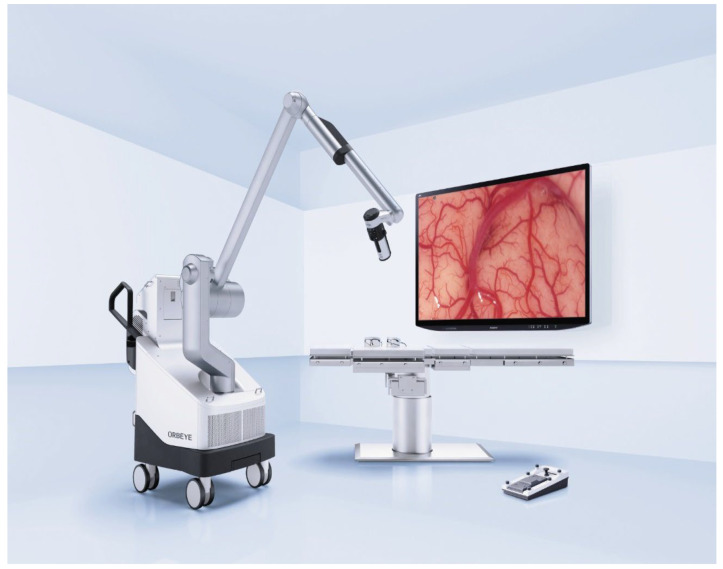
ORBEYE 4K 3D system (Olympus Inc., Tokyo, Japan), an exoscope.

**Figure 3 life-13-00692-f003:**
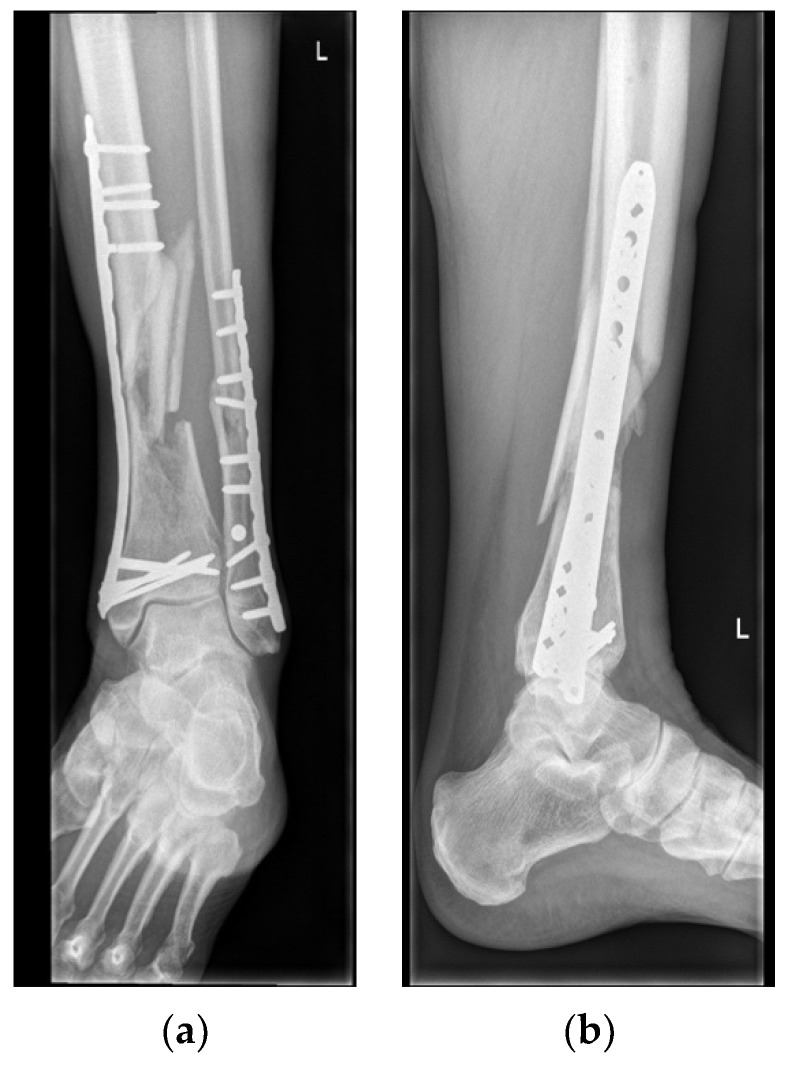
Pre-operative X-ray demonstrating nonunion of the comminuted fracture containing (necrotic) bone parts (**a**,**b**).

**Figure 4 life-13-00692-f004:**
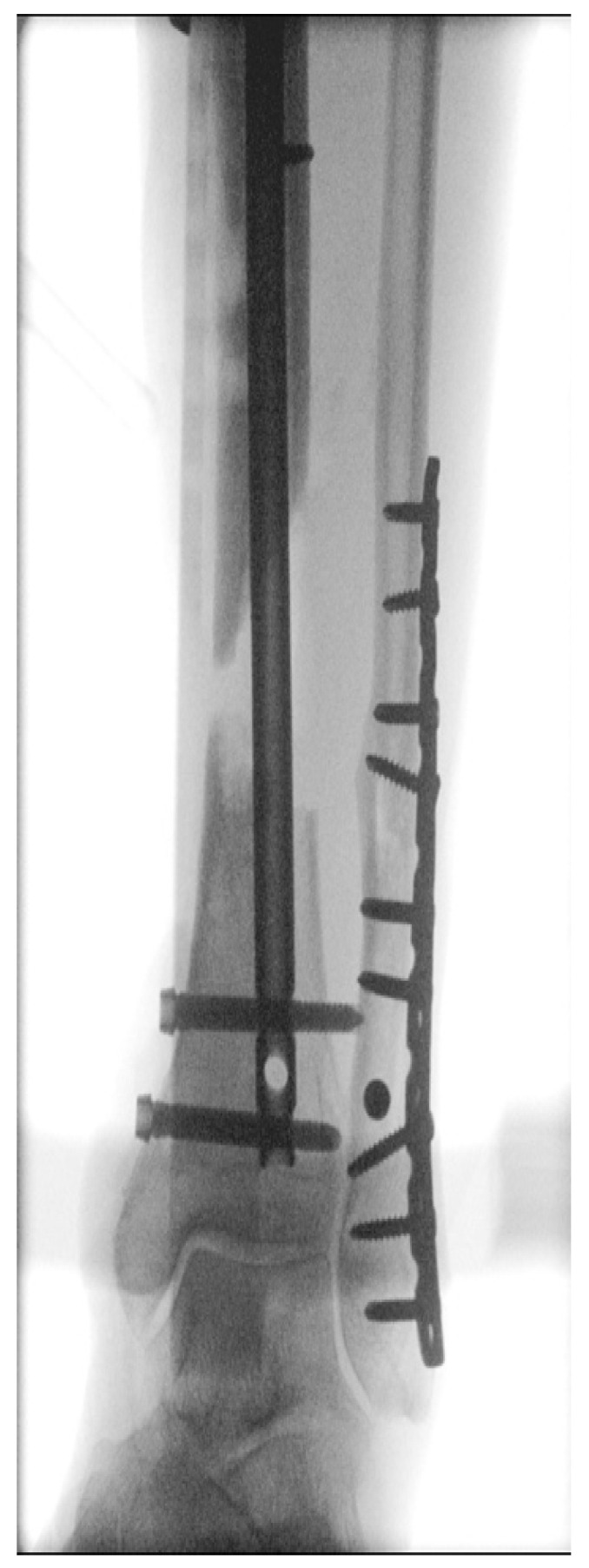
Intraoperative X-ray: debrided necrotic bone and soft tissue and placement of cement spacer and T2 tibial nail.

**Figure 5 life-13-00692-f005:**
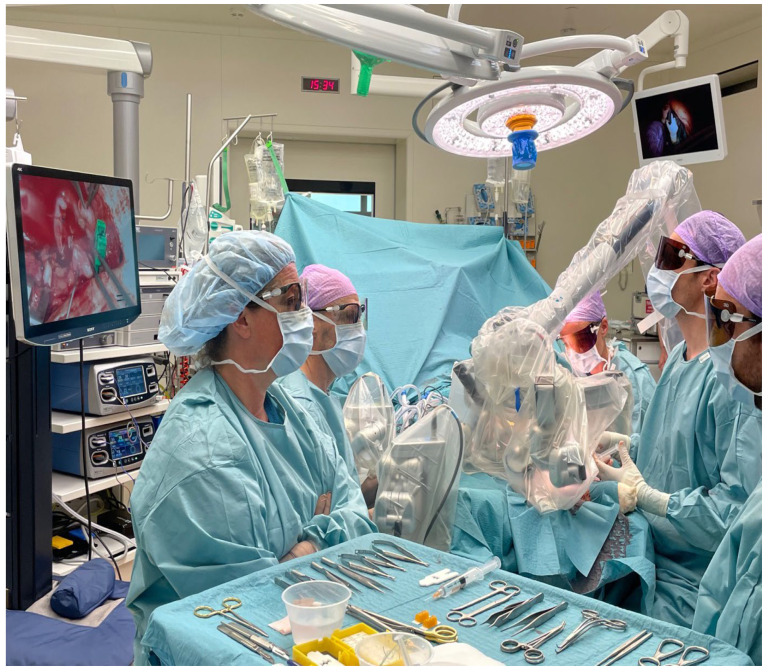
MUSA robot and ORBEYE system draped and installed above surgical field.

**Figure 6 life-13-00692-f006:**
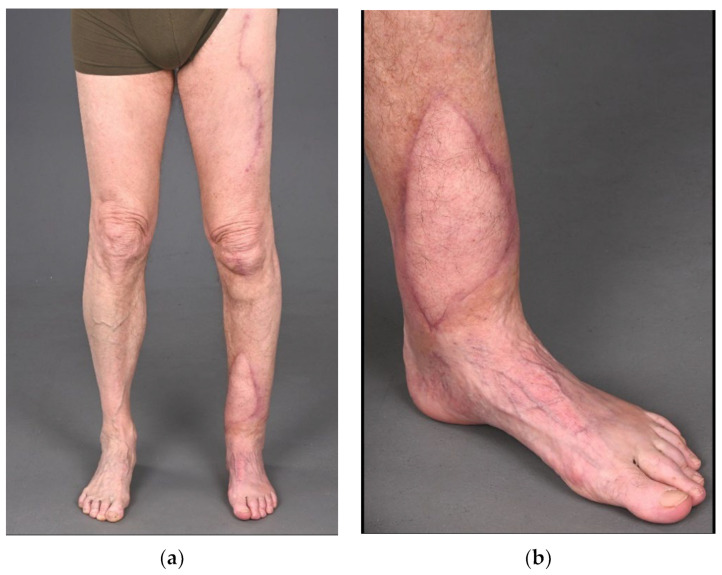
Clinical photos of the ALT flap on the lower leg and donor site of the upper leg demonstrating a vital coverage of the bone reconstruction (**a**,**b**). (21 December 2022).

**Figure 7 life-13-00692-f007:**
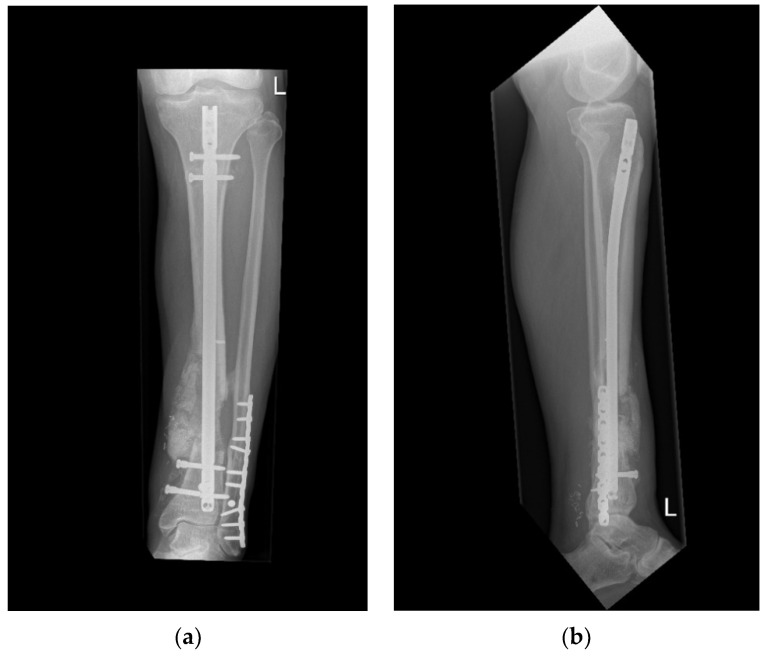
X-ray 4 months postoperatively demonstrating progressing consolidation (**a**,**b**). (December 2022).

## Data Availability

Not applicable.

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
