# Peer review of "First-in-Human Integrated Use of a Dedicated Microsurgical Robot with a 4K 3D Exoscope: The Future of Microsurgery"

_life, 2023, doi:10.3390/life13030692_

Round 1

Author Response

The authors are grateful for the remarks and are indebted to the reviewer for carefully reading their manuscript and providing detailed comments. A point-by-point feedback by the authors is given below. The changes in the text are high-lighted as ‘track-changes’

Reviewer 1

The somewhat bold and overly enthusiastic style of the manuscript has the potential to motivate further research and attract investors in the field. While this may feel a bit unconventional for a scientfic paper, I am inclined to leave it just the way the authors have written it and trust that it will fit into the scope of the journal.

Two minor things I would recommend to change on page 8:

The authors state that: Currently the camera systems of available exoscopes do not yet have the quality of operation microscopes when very high magnification is required, but this evolution will continue and these camera systems will probably ultimately replace the current operation microscopes.

This statement does not hold true for the current flagship systems of Zeiss, Leica, Aesculap and Arriscope anymore. These optics are comparable to analog systems.
The Orb Eye is special, since it offers a compromise by introducing a very easily employable digital microscope with a limited resolution. While it does not provide high-end optics, it ledns itself excellently to a combination with a microsurgical robot like MUSA. These thoughts should replace the current statement.

Author reply: we thank you for your comment, we modified this paragraph and included a nuance in the discussion section.

Numerous other innovations have the ability to propel our possibilities in medicine. Besides the incorporation of other imaging techniques enhancing our eyes and robots en-hancing our hands, 3D navigation, augmented/virtual reality technology and artificial in-telligence can help surgeons to have a better understanding of the anatomy and enable more precision, decreasing donor morbidity. We should continue to explore and develop these potential benefits in a structured and judicious way considering safety, costs and limitations and continue the path to better care for our current and future patients.

This paragraph should be omitted since it does not add to the conclusion of the paper and also is not based on the results of the case at hand.

Author reply: this paragraph has been deleted. The last “general” sentence has been placed at the end of the conclusion.

Reviewer 2 Report

Regarding the paper:

First-in-human integrated use of a dedicated microsurgical robot with a 4K 3D exoscope: the future of microsurgery

Review:
Abstract
Good and well and fits te required explanations.

Introduction

Would like more citations of robotic evolutions and its positive impacts on microsurgery and supermicrosurgery, easy learning curve, etc

Material and Methods Good

Results Good

Discussion:
Need to reinforce drawn backs such as cost and time to docking the robot. If there are other limitations of the method please mention.

Overall
Acceptable for publiction

Author Response

The authors are grateful for the remarks and are indebted to the reviewer for carefully reading the manuscript and providing detailed comments. A point-by-point feedback by the authors is given below. The changes in the text are high-lighted as ‘track-changes’

Reviewer 2

Regarding the paper:

First-in-human integrated use of a dedicated microsurgical robot with a 4K 3D exoscope: the future of microsurgery

Review:
Abstract
Good and well and fits te required explanations.

Introduction

Would like more citations of robotic evolutions and its positive impacts on microsurgery and supermicrosurgery, easy learning curve, etc

Material and Methods Good

Results Good

Discussion:
Need to reinforce drawn backs such as cost and time to docking the robot.

Overall
Acceptable for publiction

Author reply: we thank you for your positive comments. We included additional information and citations to the introduction and discussion section regarding robotic evolutions, positive impacts, learning curves and draw backs such as docking and costs.

Reviewer 3 Report

Excellent report for the use of Exoscopic microsurgery along with the novel robot. 

Proof of concept for the utilisation of both the exoscope and the robot in a challenging microsurgical reconstructive case.

Author Response

The authors are grateful for the remarks and are indebted to the reviewer for carefully reading the manuscript and providing detailed comments. A point-by-point feedback by the authors is given below. The changes in the text are high-lighted as ‘track-changes’

Reviewer 3

Excellent report for the use of Exoscopic microsurgery along with the novel robot. 

Proof of concept for the utilisation of both the exoscope and the robot in a challenging microsurgical reconstructive case.

Author reply: we thank you for your positive comments.
